# Marine Bacterial Polysaccharide EPS11 Inhibits Cancer Cell Growth and Metastasis via Blocking Cell Adhesion and Attenuating Filiform Structure Formation

**DOI:** 10.3390/md17010050

**Published:** 2019-01-11

**Authors:** Ju Wang, Ge Liu, Weiping Ma, Zhongxia Lu, Chaomin Sun

**Affiliations:** 1CAS Key Laboratory of Experimental Marine Biology, Institute of Oceanology, Chinese Academy of Sciences, Qingdao 266071, China; 18317898332@163.com (J.W.); liug878@163.com (G.L.); 2Laboratory for Marine Biology and Biotechnology, Qingdao National Laboratory for Marine Science and Technology, Qingdao 266071, China; 3Department of Earth Science, University of Chinese Academy of Sciences, Beijing 100049, China; 4Center for Ocean Mega-Science, Chinese Academy of Sciences, Qingdao 266071, China; 5School of Medicine and Pharmacy, Ocean University of China, Qingdao 266003, China; maweiping1990@163.com (W.M.); 18158507989@163.com (Z.L.)

**Keywords:** polysaccharide, EPS11, cancer, filiform structure, adhesion, CD99, metastasis

## Abstract

Our previous results suggested that EPS11, a novel marine bacterial polysaccharide, might be a potential drug candidate for human non-small cell lung carcinoma treatment. In this study, we further investigate the anticancer mechanisms against liver cancer and the anti-metastatic effects in vivo of EPS11. Firstly, we found that EPS11 exerts cytotoxic effects via blocking cell adhesion and destroying filiform structure formation in Huh7.5 cells. Moreover, mass spectrometry-based proteomic analysis of EPS11-treated Huh7.5 cells revealed that expression of many adhesion-related proteins was significantly changed. It is noteworthy that the expression of CD99, a key factor related to cell adhesion, migration and cell death, is remarkably down-regulated after EPS11 treatment. Importantly, over-expression of CD99 partly rescues cell death rate, and improves cell adhesion and migration ability in Huh7.5 treated by EPS11. Thus, we propose that CD99 is a potential action target of EPS11, inhibiting cancer cell proliferation, adhesion and migration. Notably, administration of EPS11 simultaneously with tumor induction evidently reduces tumor nodule formation in the lungs, which strongly indicates that EPS11 has anti-metastatic effects in vivo. Taken together, our results suggest that EPS11 inhibits liver cancer cell growth via blocking cell adhesion and attenuating filiform structure formation, and has potential as an anti-cancer drug, targeting metastasis of cancer cells, in the future.

## 1. Introduction

Hepatocellular carcinoma (HCC) is the most common liver cancer, accounting for approximately 90% [1,2], and it has a five-year survival rate around only 30–40% [3]. Similar to other kinds of cancers, tumor metastasis remains the primary cause of mortality of HCC patients [4]. It is well known that metastasis is a multistage process that requires cancer cell detachment from the primary tumor, survival in the circulation, settlement at distant sites and growth [5]. During this process, metastatic cells go through detachment, migration, invasion and adhesion, and inhibition of any one of these steps has the potential to block the entire metastatic process [6]. However, there is very little advancement in the area of blocking tumor metastasis [6], therefore, new effective agents possessing anti-metastasis activities are urgently needed for better prevention and treatment of HCC and other cancers.

Recently, marine microbial polysaccharide has become an important resource for searching anti-metastasis agents because they possess numerous advantages including biocompatibility, biodegradability, nontoxicity, low cost and abundance [7]. Heymann and colleagues found that a marine bacterial exopolysaccharide, OS-EPS, was potent in inhibiting both migration and invasiveness of osteosarcoma cell lines and was very efficient in inhibiting the establishment of lung metastases in vivo [8]. In our previous study, we found that a novel marine bacterial polysaccharide, EPS11, showed strong cytotoxic effects on lung cancer cells through affecting cell adhesion and anoikis [9]. Notably, filiform structures of A549 cells were markedly destroyed by EPS11 in a dose-dependent manner [9]. Filiform structures include filopodia, invadopodia, filopodium-like protrusions and actin spikes [10]. There is increasing evidence showing that cancer cells use filiform structures, in cell adhesion and three-dimensional cell migration, both in vitro and in vivo [10]. Metastatic tumors cells are rich in filiform structures, and the number of which correlates with their invasiveness [11]. Recently, filiform structures have also been implicated as critical for metastatic tumors cells to colonize in secondary tissues or organs [12]. Hence, the formation of filiform structures represents a critical rate-limiting step for the subsequent development of macroscopic metastases and becomes a hot target for anti-cancer drug development.

Meanwhile, lots of studies have stated that many molecular factors are involved in the process of HCC metastasis, including adhesion molecules, matrix metalloproteinases, vascular growth factors, oncogenes, etc. [13]. CD99, a glycosylated transmembrane protein, has been reported to have a marked effect on the migration, invasion, and metastasis of tumor cells through multiple and unclear action mechanisms, thereby emerging as a novel therapeutic target [14]. However, much still needs to be understood. Nevertheless, several studies indicated that agonistic CD99 monoclonal antibodies were able to activate cell death signals and inhibit cell migration [14]. Furthermore, CD99 is positively stained in immunohistochemical preparations of HCC tissue material and negatively stained in non-HCC tissues outside the liver or metastatic within the liver [15], which strongly suggests that CD99 is essential in the course of cancer development and metastasis of HCC.

Based on our previous results, EPS11 preferentially killed cancer cells, including human lung cancer cells, A549, and HCV-related human liver cancer cells, Huh7.5, compared with normal cell line human embryonic lung fibroblasts, WI-38, and potentially possessed anti-metastasis activity [9]. However, the detailed molecular mechanisms of EPS11 against liver cancer and its potential anti-metastasis activity are not clear. In this work, we first investigated the suppression of EPS11 on Huh7.5 cells’ filiform structure formation, adhesion and migration abilities. Then we chose CD99 as a reference for anti-cancer mechanism studies of EPS11 based on proteomic analysis. Finally, we verified the inhibition activity of EPS11 against metastasis in a melanoma mouse model.

## 2. Results

### 2.1. Cytotoxic Effects of EPS11 on Huh7.5 Cells

The purification of EPS11 was performed by the procedures described in Material and Methods. The cytotoxic effects of corresponding eluted components on Huh7.5 cells were determined by 3-(4,5-Dimethylthiazol-2-yl)-2,5-diphenyltetrazolium bromide (MTT) assay. The results showed that the polysaccharide content of eluted components was positively associated with the cytotoxic effect on Huh7.5 cells (Figure 1A), which is very similar to the previous results of EPS11 against A549 cells [9]. The relative viabilities of Huh7.5 cells were further determined when treated with different concentrations of EPS11 (0–180 nM) for 24 hours and 48 hours. As expected, EPS11 significantly inhibited the growth of Huh7.5 cells in time- and dose-dependent manners (Figure 1B). Notably, the growth inhibition rates of EPS11 against the other two liver cancer cell lines HepG2 and 7402 were very similar to that in Huh7.5 (Appendix A).

### 2.2. EPS11 Suppressed Cell Adhesion, Filiform Structure Formation and Cell Migration in Huh7.5 Cells

In the previous study, we found that A549 cell detachment from extra cellular matrix was the most obvious and repeatable effect when treated with EPS11 [9]. Similarly, Huh7.5 cells lost adhesion capability and formed evident aggregation in a dose-dependent manner when treated with EPS11 (Figure 2A). Hence, we preformed the quantification assay via crystal violet staining to further check the adhesion ability of Huh7.5 cells after treatment with different concentrations of EPS11 (0–18 nM). As shown in Figure 2B, EPS11 significantly decreased the number of adhered Huh7.5 cells in time- and dose-dependent manners. When the concentration of EPS11 increased to 3.6 nM, almost all the cells were detached from the extra cellular matrix after 24 hours incubation. Additionally, we investigated the cell adhesion rate in the other two liver cancer cell lines, HepG2 and 7402, in the presence of different concentrations of EPS11. Consistently, the cell adhesion rates in both cell lines, HepG2 and 7402, were evidently suppressed when treated with different concentrations of EPS11 (Appendix A). Notably, human hepatoma Huh7.5 cell line is closely associated with hepatitis C virus-related human liver cancer, and this kind of liver cancer is becoming more and more serious in the world. Thus, we chose Huh7.5 as our model to investigate the anti-cancer mechanisms of EPS11.

To further disclose the effects of EPS11 on Huh7.5 cell surface membrane structures, we observed Huh7.5 cells treated with different concentrations of EPS11 (0–9 nM) by scanning electron microscope (SEM). As shown in Figure 2C, Huh7.5 cells in the control group showed regular adherent growth with long and multiple filiform structures (Figure 2C, 0 nM treatment), which play essential roles in cell adhesion. Notably, the numbers of filiform structures significantly decreased along with the increase in EPS11 concentration (Figure 2C). In addition, the cells shifted to a round shape and lost almost all filiform structures at the concentration of 9.00 nM (Figure 2C, 9.00 nM treatment). The inhibition tendency of filiform structure formation is very consistent with what we observed in the cell adhesion assay (Figure 2B,C), which is very similar to those results tested in A549 cells as described previously [9].

Filiform structure is a key factor determining cell adhesion and migration in cancer cells [6]. EPS11 could effectively attenuate the formation of filiform structures and decrease the adhesion ability in Huh7.5 cells. We next sought to check whether EPS11 could inhibit the migration of Huh7.5 cells. Therefore, we examined the migration ability of Huh7.5 cells in the absence or presence of EPS11 via wound healing assay and Transwell Boyden chamber assay. The wound healing assay results showed that EPS11 could effectively reduce cell migration distance in Huh7.5 cells after the treatment EPS11 for 8 hours at the concentration of 1.8 nM (Figure 3A). For the transwell boyden chamber assay, as shown in Figure 3B, the migration ability of Huh7.5 cells was significantly weakened after the treatment of EPS11 at the concentration of 9 nM compared with the untreated group. Altogether, EPS11 could effectively suppress cell adhesion, filiform structure formation and cell migration in Huh7.5 cells.

### 2.3. Proteomic Analysis of Differentially Expressed Proteins Associated with Cell Adhesion After EPS11 Treatment

To better understand the detachment effects of EPS11 on Huh7.5 cells, we performed a proteomic analysis to identify differentially expressed proteins after EPS11 treatment. We treated Huh7.5 cells with 45 nM and 90 nM EPS11 for 24 hours and found that 765 and 362 proteins, respectively, were differentially expressed (1.5-fold change cutoff and *p* value less than 0.05). All of these differently expressed proteins were screened to identify adhesion-associated proteins, and 22 proteins were filtered and clustered (Figure 4). Among the significantly down-regulated cell adhesion-associated proteins, claudin-1 and claudin-6, together with occludin, are the major constituents of the tight junction complex [16]. Junctional adhesion molecule A is a tight junction-associated immunoglobulin superfamily protein implicated in the regulation of tight junctions and leukocyte transmigration [17]. Vitronectin, cadherin 2, laminin and fibronectin are extracellular matrix (ECM) proteins associated with cell adhesion and migration [18,19,20,21]. Nidogen-1 and -2 are key components of basement membranes [22]. Matrix metalloproteinase-14 (MMP14) is a transmembrane protein enhancing tumor growth and invasion [23]. The intercellular cell adhesion molecule 1 (ICAM-1) is a transmembrane molecule that participates in many important processes, including leukocyte endothelial transmigration, cell signaling, cell-cell interaction, cell polarity and tissue stability [24]. Notably, the expression of CD99, a cell surface molecule, was significantly reduced. It gained our attention because of its involvement in regulating cell differentiation and adhesion/migration of immune and tumor cells [14]. Thus, we chose CD99 as a clue to investigate the effects of EPS11 on the growth and adhesion inhibition in Huh7.5 cells in the following study.

### 2.4. EPS11 Down-Regulated the Expression of CD99

Considering the importance of CD99 in regards to cell adhesion, migration and cancer metastasis and its significant down-regulation by EPS11 based on the proteomic result, we explored the expression of CD99 in both mRNA and protein levels after treatment with EPS11. The result of quantitative reverse transcription-PCR (qRT-PCR) showed that EPS11 could dose-dependently decrease the transcriptional level of CD99 mRNA (Figure 5A). Consistently, the protein level of CD99 was also reduced in a concentration-dependent manner after treatment with EPS11 (Figure 5B). However, the expression of CD99 decreased dramatically after treatment of EPS11 at a higher concentration, and no obvious expression could be detected after 6 hours with the treatment of 4.5 nM EPS11 (Figure 5C).

### 2.5. The Over-Expression of CD99 Rescued EPS11-Induced Suppression of Cell Viability, Adhesion and Migration in Huh7.5 Cells

To further evaluate the functional importance of the EPS11-mediated inhibition of CD99, the CD99 encoding gene was cloned and over-expressed in Huh7.5 cells through transfection. As shown in Figure 6A, the expression levels of CD99 in Huh7.5 cells were dramatically increased after transfection for 12 hours and 24 hours compared with the control group. CD99 plays essential roles in the course of cell adhesion, migration and metastasis, therefore, we tested whether the over-expression of CD99 could rescue the EPS11-induced cytotoxicity on Huh7.5 cells. As shown in Figure 6B, the over-expression of CD99 could significantly restore the cell viability in Huh7.5 cells after EPS11 treatment with different concentrations (0–22.5 nM) for 24 hours, especially for the concentrations of EPS11 at 0.09 and 0.45 nM. We further examined whether the over-expression of CD99 could rescue the EPS11-induced suppression of cell adhesion and cell migration abilities. As expected, the decreased cell viability, cell adhesion and cell migration in EPS11-treated Huh7.5 cells were all restored by the over-expression of CD99 (Figure 7A–C). These results suggest that the reduced expression of CD99 led to the suppression of cell viability, adhesion and migration.

### 2.6. EPS11 Suppressed Melanoma Metastasis in the Mice Model

Based on our results, EPS11 effectively suppressed cell adhesion, filiform structure formation and cell migration in Huh7.5 cells, which strongly suggests that EPS11 might be an inhibitor of cancer cell metastasis. Therefore, we evaluated the anti-metastasis effects of EPS11 by pulmonary colonization assay. Melanoma is one of the most aggressive forms of skin cancer, and is a typical metastasis model to test the effect of anti-metastasis drugs [25,26,27,28]. Therefore, in this study, we chose the highly metastatic human melanoma cell lines B16F-10 to verify the anti-metastasis effect of EPS11 in vivo. Notably, melanoma metastatic tumor-bearing animals treated with EPS11 showed a significant reduction in tumor nodule formation, while metastatic animals in the control group had a massive melanoma tumor nodule compared with the EPS11 treated group (Figure 8A). Meanwhile, the mice weight had the same tendency for both the EPS11 treated group and the control group (Figure 8B). Hence, 200 mg/kg daily EPS11 dosing in mice over a 9-day period did not result in any toxicity that was specifically attributed to EPS11. Together, these data demonstrate that EPS11 can markedly decrease the lung colonization of melanoma tumor cells in animal models.

## 3. Discussion

Numerous studies have shown that marine bacterial polysaccharides with novel chemical compositions, properties and structures have been found to have potential applications in pharmaceuticals and medicine as anti-cancer drugs, food additives and so on [29]. EPS11, a novel marine bacterial polysaccharide, preferentially killed cancer cells including human lung cancer cells, A549, and HCV-related human liver cancer cells, Huh7.5, compared with normal cell line human embryonic lung fibroblasts, WI-38 [9]. EPS11 was further demonstrated to inhibit lung cancer cell growth via blocking filopodia mediated adhesion and stimulating βIII-tubulin associated anoikis [9]. However, the detailed molecular mechanisms against liver cancer and its potential anti-metastasis effect are not clear.

In this study, we first demonstrated that EPS11 inhibited cell growth and adhesion by destroying filiform structures in the liver cancer cell line, Huh7.5 (Figure 2A,C), which is very similar to the effects on lung cancer cells treated with EPS11 [9]. Filiform structures are thin, finger-like and highly dynamic actin-rich membrane protrusions that extend out from the cell edge and play a central role in modulating cell adhesion, migration and cancer cell invasion [10]. Increased numbers of filiform structures have been implicated in the formation and reinforcement of cell-cell junctions, cell migration and invasion [6]. In our study, the results showed that EPS11 destroyed Huh7.5 cell filiform structures in a dose-dependent manner (Figure 2C). Meanwhile, destroying filiform structures has been associated with a reduced adhesion rate (Figure 2B) and migration ability (Figure 3A, B). Thus, EPS11 could inhibit Huh7.5 cell adhesion and migration ability by destroying filiform structures. Consistently, EPS11 also significantly suppressed the formation of filiform structures in three other lung cancer cell lines including A549, H1299 and H460 [9], which strongly suggests that filiform structure is one of the key action targets of EPS11.

Presently, proteomic technologies not only provide a powerful tool to profile protein expression in response to various stresses in cancer cells, but are also used to reveal the possible mechanism of action of corresponding drugs [30]. Based on the proteomic results of Huh7.5 treated with EPS11, we could see clearly that many proteins associated with cell adhesion, migration and metastasis were evidently down-regulated, which coincided with the results related to inhibition of cell adhesion and migration in Huh7.5 shown above (Figure 2 and Figure 3). Among these proteins, CD99 is reported to be involved in regulating tumor growth, differentiation, cell adhesion, cell migration and metastasis [14]. High CD99 expression plays an oncogenetic role in Ewing sarcoma, lymphoblastic lymphoma, myeloid chondrosarcoma, malignant glioma and so on [31,32,33,34]. It is noteworthy that CD99 has a positive expression in hepatocellular carcinoma, but in non-HCC cancer, it has a negative expression [15,35]. Despite increasing evidence that CD99 has important functions in several aspects of cell biology, this molecule has been largely ignored by the scientific community. A number of unresolved issues remain to be clarified, particularly in terms of the mechanisms of action of CD99. In this study, EPS11 treatment significantly reduced CD99 expression in both mRNA and protein levels (Figure 5A,B), which is consistent with the proteomic result (Figure 4). Furthermore, the over-expression of CD99 in Huh7.5 cells partly prevented the EPS11-mediated cell death, loss of adhesion rate and reduction of cell migration ability (Figure 6B and Figure 7A–C). The results suggested that the down-regulation of CD99 expression by EPS11 was partly responsible for the suppression of cell growth, cell adhesion and migration, and filiform structure formation mentioned above. In our previous results, EPS11 inhibited the growth of human non-small cell lung carcinoma via blocking filiform structure-mediated adhesion and stimulating βIII-tubulin-associated anoikis [9]. However, the expression of βIII-tubulin in Huh7.5 was not evidently affected by EPS11, which indicates that EPS11 might inhibit cancer cell growth with slightly different mechanisms, even though it apparently attenuated different cell lines’ adhesion and led cells to aggregate.

Moreover, EPS11 significantly inhibited cell adhesion and migration, which are necessary for tumor metastasis. Metastasis is a complex, multistep process, during which tumor cells spread from the primary tumor mass to distant organs [36]. The ability of cancer cells to disseminate from the primary site and form distant metastases is the main cause for cancer-related morbidity in patients with solid tumors. Notably, the melanoma tumor nodules in lung were significantly reduced after EPS11 treatment (Figure 8A). Combining our previous and current results in different cancer cell lines, we conclude that EPS11 has potential to inhibit several key steps of metastasis, including cell adhesion, migration and filiform structure formation. Thus, it is reasonable to develop anti-cancer drugs targeting metastasis with EPS11 in the future. Moreover, our results found that several marine bacterial polysaccharides derived from costal and deep sea also dramatically attenuated the cell adhesion and migration and the formation of filiform structures in Huh7.5 (results not shown). Therefore, we believe that marine bacterial polysaccharides are a valuable resource to develop drugs blocking tumor metastasis in the future. However, there is still work needed to be done, such as the detailed mechanisms suppressing filiform structure formation, the exact action target and the elucidation of the structure of EPS11.

In summary, we have shown that EPS11 inhibited Huh7.5 cell growth and adhesion in vitro by destroying filiform structure and down-regulating the expression of CD99. Meanwhile, EPS11 had a significant anti-metastasis effect on melanoma in animal experiments. Thus, combining the anti-cancer effects of EPS11 towards human non-small cell lung carcinoma, we further confirmed that EPS11 should be a promising lead compound for novel anti-cancer drug development.

## 4. Materials and Methods

### 4.1. Cell Culture, Reagents and Antibodies

Huh7.5, B16F-10 cell lines were obtained from the American Type Culture Collection (Manassas, Virginia, USA), and they were cultured in RPMI 1640 medium supplemented with 10% FBS (all of them purchased from Gibico, Grand Island, NY, USA), penicillin at 100 units/mL, and streptomycin at 100 μg/mL (HyClone, Logan, UT, USA) in a humidified atmosphere of 5% CO_2_ at 37 °C. MTT, BCA kit for protein quantification were purchased from Beyotime Institute of Biotechnology (Shanghai, China), and the enhanced chemiluminescence (ECL) was purchased from Pierce (Thermo Scientific, Portsmouth, NH, USA). Anti-CD99 monoclonal was purchased from KleanAB (Sangon Biotech, Shanghai, China). Antibody against β-actin was obtained from Proteintech (Wuhan, China).

### 4.2. Purification and Anticancer Activity Assay of EPS11

The purification of EPS11 was carried out as previously described with minor modifications [9]. Briefly, marine bacterium *Bacillus* sp. 11 was cultured in 2216E medium (5 g/L tryptone, 1 g/L yeast extract, 1 L filtered seawater, pH adjusted to 7.4–7.6) supplemented with 1% sucrose at 150 rpm for 2 days at 28 °C. The culture supernatant was concentrated by centrifugation at a speed of 8000 rpm at 20 °C for 20 minutes, and the polysaccharide was precipitated with three volumes of 95% (v/v) ethanol. After keeping at 4 °C overnight, the precipitate was collected by centrifugation with a speed of 8000 rpm at 4 °C for 20 minutes and dissolved in distilled water. After proteins in the samples were removed by sevage reagent (chloroform/n-butyl alcohol = 5:1; v/v), the solution was dialyzed against ddH_2_O with an 8000–14,000 molecular weight cutoff membrane. The crude polysaccharide was dissolved in ddH_2_O and loaded onto a 5 mL HiTrapTM Q HP column (GE Healthcare, Little Chalfont, UK) equilibrated with start buffer (20 mM Tris–HCl, pH 8.0), then collected with eluent (2 M NaCl in 20 mM Tris–HCl, pH 8.0). The retained fraction was dialyzed and concentrated by lyophilized. To get high purity polysaccharide, the crude polysaccharide was further purified using HiloadTM 16/600 SephadexTM 200 column (GE Healthcare, Little Chalfont, UK) pre-equilibrated with 50 mM NaCl in 20 mM Tris-HCl (pH 9.0). The fractions were collected for 4 mL/tube, and the content of polysaccharide was analyzed by phenol-sulfuric acid assay. In phenol-sulfuric acid assay, 0.5 mL solutions from each fraction in the gel filtration were mixed with 0.5 mL 5% phenol respectively, and then 2.5 mL 95.5% sulfuric acid was added. After mixing and cooling down for 20 minutes at 20 °C, the polysaccharide content was determined at OD490 nm. The corresponding ability inhibiting Huh7.5 cells growth was detected by MTT assay described as following.

### 4.3. Cell Proliferation Viability Assay

Viabilities of Huh7.5 cells were measured by the MTT method. Briefly, exponentially growing Huh7.5 cells (5 × 10^4^ cell/mL) were routinely seeded into 96-well plate at 37 °C for overnight. Then the cells were treated with different concentrations of EPS11 (0–180 nM) for 24 hours and 48 hours. 30 μL of 5 mg/mL MTT (Sigma, St. Louis, MO, USA) was added into each well and incubated for 3 hours at 37 °C, and 100 μL “Triplex Solution” (10% SDS-5% isobutanol-12 mM HCl) was added to each well. After incubation overnight, the absorbance of each well was measured at a wavelength of 570 nm by a multi-detection microplate reader (Infinite M1000 Pro, TECAN, Mannedorf, Switzerland). Relative cell viability was presented as a percentage relative to the control group. All experiments were performed three times.

### 4.4. Scanning Electron Microscope (SEM)

Huh7.5 cells were plated onto laminin-coated glass coverslips overnight and treated with different concentrations of EPS11 (0–9.0 nM). After incubation for 6 hours, the glass coverslips were washed with PBS and fixed with 5.0% glutaradehyde in PBS and gradually dehydrated in ethanol (30%, 50%, 70%, 90% and 100% for 10 minutes at each step). Finally, the glass coverslips were observed and imaged by SEM (Hitachi S-3400N, Tokyo, Japan).

### 4.5. Cell Adhesion Assay

Huh7.5 cells (5 × 10^4^/mL) were seeded into a 96-well plate at 37 °C overnight and then treated with varying concentrations of EPS11 (0–18 nM) for 12 hours and 24 hours. Cell culture medium and the non-adherent cells were discarded and washed three times with PBS, fixed with 95% ethanol for 30 minutes, and then stained with 100 μL 0.1% crystal violet for 20 minutes. While detached cells were washed off, only adhesive cells could be stained. After removing the redundant crystal violet, 100 μL acetic acid was added into each well with gentle shaking for 10 minutes. The absorbance was measured by a multi-detection microplate reader (Infinite M1000 Pro, TECAN, Mannedorf, Switzerland). Relative adhered cells were presented as a percentage relative to the control group. All experiments were performed three times.

### 4.6. Wound Healing Migration Assay

Huh7.5 cells migration ability treated with EPS11 was measured by wound healing assay. Briefly, Huh7.5 cells were incubated in serum-free medium overnight for synchronization and then trypsinized from culture dishes and placed into 12-well plate with a Culture-Insert 2 well. After growth to 90% confluence, the Culture-Insert 2 well was gently removed by sterile tweezers. After washing three times with PBS, the cells were exposed to a medium supplemented with 2% FBS with or without EPS11 (0, 0.9 and 1.8 nM). After incubation for 24 hours, three fields of each wound were selected and photographed with an inverted microscope (NIKON TS100, Tokyo, Japan) equipped with a digital camera.

### 4.7. Transwell Migration Assay

The migration ability of Huh7.5 cells was measured by a Transwell Boyden chamber (Costar; Corning Life Sciences, Lowell, MA, USA). Briefly, Huh7.5 cells were cultured in serum–free medium RPMI 1640 for overnight, then the cells were suspended in a 200 μL medium containing different concentrations of EPS11 (0–9.0 nM) that was added into the upper compartment, then 600 μL medium supplemented with 20% FBS was added into the lower compartment. After incubation at 37 °C for 8 hours, cell culture medium was discarded and the upper surface cells were gently removed by cotton swabs. After washing with PBS, cells were fixed with 95% ethanol and stained with 0.1% crystal violet. Then the non-migrated cells on the upper side of the filter were gently removed using cotton swabs and the migrated cells on the lower side of the filter were observed and counted in five random fields.

### 4.8. Proteomic Analysis

Proteomic analysis of Huh7.5 cells treated with EPS11 was performed by PTM Biolabs Co., Ltd. (HangZhou, Zhejiang, China). Briefly, Huh7.5 cells were treated with different concentrations of EPS11 (0, 45, 90 nM) for 24 hours, and cells were collected and lysed to obtain total cellular protein. Protein samples were then digested, labeled, separated and quantified by LC-ESI-MS/MS analysis. The bioinformatic analyses of protein annotation, functional classification, functional enrichment and cluster analyses were then performed as described previously [37].

### 4.9. Quantitative Reverse Transcription-PCR (qRT-PCR)

Huh7.5 cells were treated with different concentrations of EPS11 (0–3.6 nM) for 12 hours. Total cellular RNA was extracted using the TRIpure reagent (Aidlad, Beijing, China) according to the manufacturer’s instructions. Quality and quantity of RNA samples were determined by NanoDrop analysis (NanoDrop 2000, Thermo, Beijing, China). RNA was reverse transcribed into cDNA by using a Prime Script^®^ RT reagent kit (Takara Biotechnology Co., Ltd., Dalian, China). Then qRT-PCR was carried out with SYBR premix real-time PCR Reagents (Takara Biotechnology Co, Ltd., Dalian, China) by an ABI7900 real-time PCR system (Applied Biosystems, Foster City, CA, USA). Housekeeping gene β-actin was used as an internal control. The comparative 2^−ΔΔCt^ method was used to calculate the relative expression. For CD99-F1: 5′-GCCACAGGAAAGAAGGGGAA-3′, CD99-R1: 5′-CCCTTGTTC TGCATTTTCTTTGA-3′, β-actin-F: 5′-CACGATGGAGGGCCGGACTCATC-3′, β-actin-R: 5′-TAAAGACCTCTATGCCAACACAGT-3′. All qRT-PCR runs were conducted with three biological and three technical replicates.

### 4.10. Western Blot Analysis

Huh7.5 cells were treated with varying concentrations of EPS11 (0–9 nM) for different times. Cells were then collected and lysed with RIPA buffer (Sigma, St. Louis, MO, USA) with protease inhibitor PMSF (Sigma, St. Louis, MO, USA). Equal amounts of protein determined by BCA kit (Beyotime, Shanghai, China) were separated by 12% SDS-PAGE gels, and then electro-transferred to nitrocellulose membranes and incubated with primary antibodies (anti-CD99, anti-β-actin) and secondary antibodies. Finally, the results were analyzed with an ECL chemiluminescence kit and a Molecular Imager^®^ ChemiDoc™XRS system (Bio-Rad Laboratories, Inc., Pleasenton, CA, USA). Anti-β-actin was used to normalize protein loading.

### 4.11. DNA Constructs and Transfection

Total RNA of Huh7.5 cells was extracted with the TRIpure reagent (Aidlad, China) according to the manufacturer’s instructions, and then reverse transcribed into cDNA by using a Prime Script^®^ RT reagent kit (Takara Biotechnology Co, Ltd, Dalian, China). The full-length cDNA sequence of CD99 was produced by polymerase chain reaction (PCR) with the primers CD99-F2 (5′-CGGGGTACCATGGCCCGCGGGGCTGC-3′) and CD99-R2 (5′- CCGCTCGAGCTATTTCTCTAAAAGAGTACG-3′). The PCR products were purified, digested with *Kpn*I and *Xho*I, and ligated into the pmax vector (Lonza Colgne, Koln, Germany). The resulting plasmid (pmax-CD99) was verified by DNA sequencing.

The plasmids pmax and pmax-CD99 were transfected into Huh7.5 cells using the translipofectamine (Bioino, Qingdao, China) according to the manufacture’s protocol. After transfection, cells were collected at 48 hours post-transfection, and proteins were extracted. Protein samples were resolved on 12% SDS-PAGE gels, electro-transferred to nitrocellulose membranes and incubated with primary antibodies against CD99, β-actin and secondary antibodies, and finally detected by enhanced chemiluminescence.

### 4.12. Pulmonary Colonization Assay

C57BL/6 mice (7-week-old males) were purchased from Beijing Vital River Laboratory Animal Technology Co., Ltd (Beijing, China). All studies on mice were approved by IOCAS (Institute of Oceanology, Chinese Academy of Sciences) Laboratory Animal Care and Ethics Committee in accordance with the animal care and use guidelines. All animal procedures were conducted in accordance with all appropriate regulatory standards under protocol Haifajizi 2013-3 (approval date: 2013-12-09). Metastases were established by giving lateral tail vein injection of 4 × 10^5^ B16F-10 melanoma cells into mice and divided into 2 groups. Mice were simultaneously administrated with EPS11 dose at 200 mg/kg (0.9 μmol/L/kg on average) once every two days by intraperitoneal injection, while equivoluminal normal saline was injected for the control group. Drug-treatment lasted for 18 days. Body weight was measured every two days with a balance for the average value. After 18 days treatment, the mice were euthanized to excise the lungs. The lungs were used for morphological examination of melanoma metastatic tumor nodules.

### 4.13. Statistical Analysis

All data are expressed as means ± SD. Statistical analysis was performed using GraphPad’s Prism v5.0 (San Diego, USA). Differences of *p* < 0.05 were considered statistically significant (**p* < 0.05, ***p* < 0.01).

## Figures and Tables

**Figure 1 marinedrugs-17-00050-f001:**
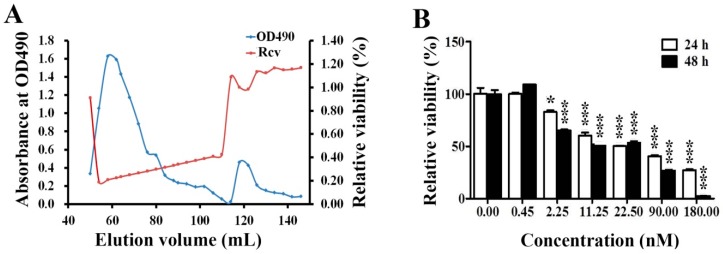
Cytotoxic effects of EPS11 on Huh7.5 cells. (**A**) Purification and activity assay of EPS11. The profiles of the fractions in the gel filtration, which were collected and monitored for the cell proliferation determined at OD570 nm after 3-(4,5-Dimethylthiazol-2-yl)-2,5-diphenyltetrazolium bromide (MTT) assay and polysaccharide content determined at OD490 nm after the phenol-sulfuric acid assay. “Rcv” stands for relative cell viability. (**B**) Cytotoxic effects of EPS11 on Huh7.5 cells. Huh7.5 cells were seeded in a 96-well plate overnight, and treated with different concentrations of EPS11 for 24 hours and 48 hours. The cell viability was analyzed by MTT assay. Data were presented as means ± SD of three independent experiments (*n* = 3). **p* < 0.05, ****p* < 0.001.

**Figure 2 marinedrugs-17-00050-f002:**
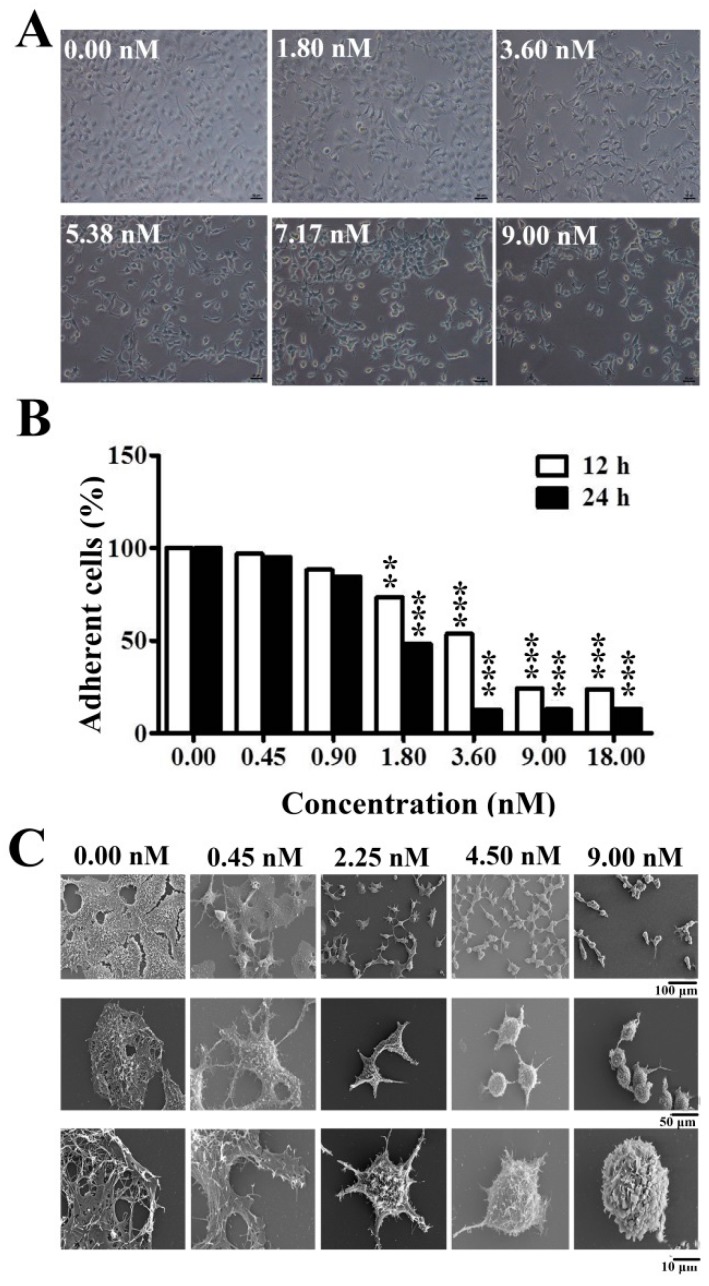
Inhibition of cell adhesion and destroying of filiform structures in Huh7.5 cells treated by EPS11. (**A**) Observation of the morphological changes in Huh7.5 cells after the treatment of different concentrations of EPS11 for 6 hours via light microscope (Nikon, Tokyo, Japan). (**B**) Quantification assay of cell adhesion in Huh7.5 after treatment with different concentrations of EPS11 for 12 hours and 24 hours. The data were presented as means ±SD of three observation fields in one representative experiment chosen from three independent experiments. **p* < 0.05, ***p* < 0.01, ****p* < 0.001. (**C**) Observation of the filiform structures in Huh7.5 cells after the treatment of different concentrations of EPS11 via scanning electron microscopy (SEM). Huh7.5 cells were treated with indicated concentration of EPS11 (0, 2.25, 4.50, 9.00 nM) for 6 hours.

**Figure 3 marinedrugs-17-00050-f003:**
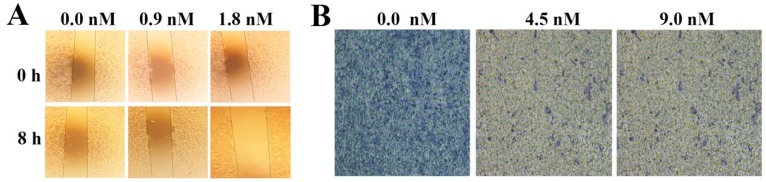
EPS11 inhibited Huh7.5 cells migration ability. (**A**) Quantitative evaluation of Huh7.5 cells migration treated with different concentrations of EPS11 through wound healing assay for 0 hours and 8 hours. (**B**) Quantitative evaluation of Huh7.5 cells migration treated with different concentrations of EPS11 through Transwell assay.

**Figure 4 marinedrugs-17-00050-f004:**
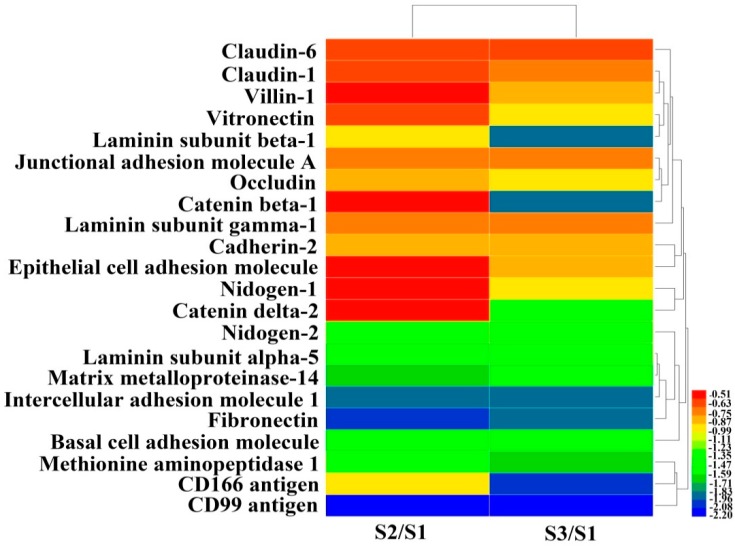
Proteomic clustering and heatmap analyses of differentially expressed adhesion-related proteins after EPS11 treatment. Huh7.5 cells were treated with different concentrations of EPS11 (S1 = 0 nM, S2 = 45 nM, S3 = 90 nM) for 24 hours and extracts from whole cell lysates were separated and identified using liquid chromatography electrospray ionisation tandem mass spectrometry (LC-ESI-MS/MS) analysis. The relative abundances of differentially expressed proteins related to cell adhesion (fold change ≤ 0.67 and *p* < 0.05) were imported for clustering analysis using HemI. Hierarchical clustering of proteins was displayed by average linkage and Euclidean distance metrics and color scales of logarithm (log^2^) values were shown. Each line represented one protein.

**Figure 5 marinedrugs-17-00050-f005:**
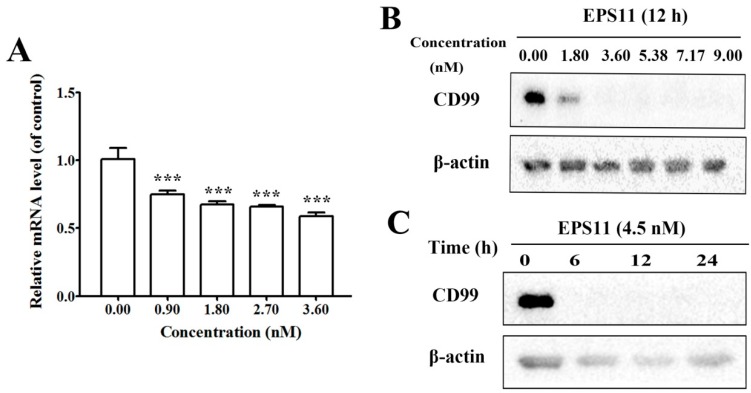
EPS11 reduced the expression of CD99. (**A**) The mRNA expression level of CD99 in Huh7.5 cells was down-regulated by EPS11. Huh7.5 cells were incubated with different concentrations of EPS11 for 12 hours, and the mRNA expression of CD99 was measured by quantitative reverse transcription-PCR (qRT-PCR). β-actin was used as the loading control. **p* < 0.05, ***p* < 0.01, ****p* < 0.001. (**B**) The protein expression level of CD99 in Huh7.5 cells was dose-dependently down-regulated by EPS11. Huh7.5 cells were incubated with different concentrations of EPS11 for 12 hours. β-actin was used as the loading control. (**C**) The protein expression level of CD99 in Huh7.5 cells was time-dependently down-regulated after treatment with 4.50 nM EPS11 for different times. The protein expression levels of CD99 were measured by Western blotting, and β-actin was used as the loading control.

**Figure 6 marinedrugs-17-00050-f006:**
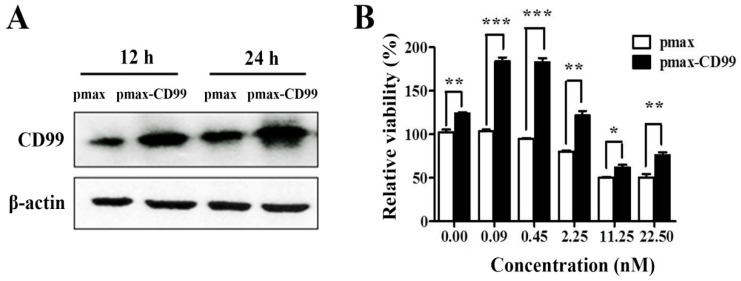
The over-expression of CD99 in the Huh7.5 cells rescued the EPS11-induced inhibition of cell viability. (**A**) The expression levels of CD99 were detected by Western blotting after Huh7.5 cells were transfected with pmax or pmax-CD99 for 12 hours and 24 hours. (**B**) The over-expression of CD99 in the Huh7.5 cells rescued the EPS11-induced suppression of cell viability. **p* < 0.05, ***p* < 0.01, ****p* < 0.001 versus control.

**Figure 7 marinedrugs-17-00050-f007:**
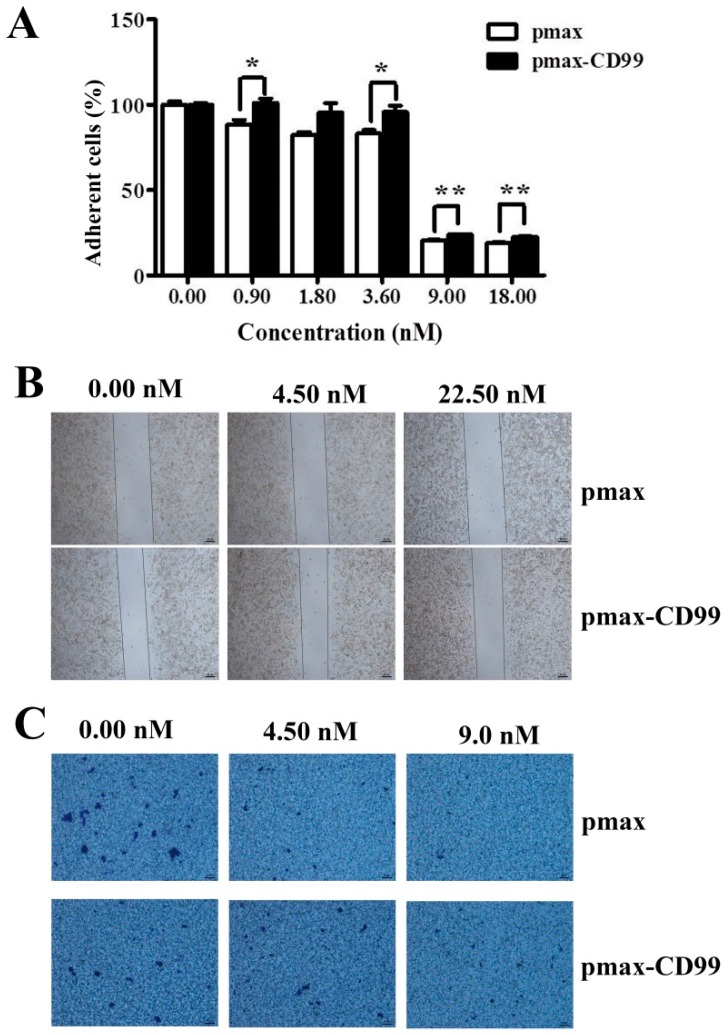
The over-expression of CD99 in the Huh7.5 cells rescued the EPS11-induced decrease in cell adhesion and migration. (**A**) The over-expression of CD99 in the Huh7.5 cells rescued the EPS11-induced reduction in cell adhesion. **p* < 0.05, ***p* < 0.01 versus control. The over-expression of CD99 in the Huh7.5 cells rescued the EPS11-induced decrease in cell migration by wound-healing assay (**B**) and Transwell assay (**C**).

**Figure 8 marinedrugs-17-00050-f008:**
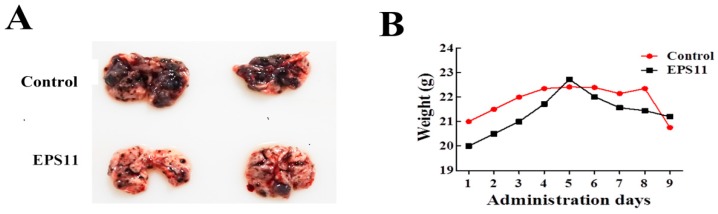
Inhibitory effect of EPS11 on melanoma tumor lung colonization. (**A**) Inhibition of C57BL/6 mouse mammary tumor cell metastasis to the lung by EPS11 in a spontaneous metastasis model. Representative images of the tumor nodule formation in the lung were observed. Mice were simultaneously administrated intraperitoneal EPS11 at a dose of 200 mg/kg (0.9 μmol/L/kg on average) and injected equivoluminal normal saline as control every two days. Drug treatment lasted for 18 days. (**B**) EPS11 had no effect on the body weights of treated mice. Body weight was measured every two days with a balance for average value.

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
