# Peer review of "Marine Bacterial Polysaccharide EPS11 Inhibits Cancer Cell Growth and Metastasis via Blocking Cell Adhesion and Attenuating Filiform Structure Formation"

_marinedrugs, 2019, doi:10.3390/md17010050_

Round 1
Reviewer 1 Report
REVIEW
Manuscript:
Marine Bacterial Polysaccharide EPS11 Inhibits Cancer Cell Growth and Metastasis via Blocking Cell Adhesion and Attenuating Filiform Structures Formation.
Dear Authors,
The authors report on EPS11 and its influence on cell growth and metastasis in cancer cells.
Due to the increasing incidence of cancers with dismal prognosis new therapies and therapy strategies are necessary.
The authors could show that the expression of CD99, a key factor related to cell adhesion, migration and cell death, is remarkably down-regulated after EPS11 treatment. They propose that CD99 is a potential action target of EPS11 inhibiting cancer cells proliferation, adhesion and migration. Furthermore anti-metastatic effects in vivo have been shown for EPS11.
Moreover, the manuscript is well-prepared and completed by excellent figures.
In conclusion, I would recommend the editors to accept the manuscript in the present form
Yours Sincerely,
Kornprat Peter, MD
Professor of Surgery
Author Response
We really appreciate the reviewer’s very positive comments for our manuscript. Please check our responses to the editor and reviewer in the attachment.

Reviewer 2 Report
The manuscirpt "Marine Bacterial Polysaccharide EPS11 Inhibits Cancer Cell Growth and Metastasis via Blocking Cell Adhesion and Attenuating Filiform Structures Formation" by Wang and colleagues is written well with interesting novel aspects on the function of ESP11 in HCC.
However, the authors only examined one HCC cell line which may not be representative for the disease. Indeed, the earlier observed effects in pulmonary carcinoma were only partly seen in HCC cell line Huh7.5. Therefore, the effects may differ between cell lines. To examine the effects in different HCC cell lines would be of high interest for further development of the drug.
For in vivo experiments the authors switched to a Melanoma model without stating the reason. In addition, only measurments of body weight were used to judge toxicity. Here, the effect on the different organ weights and structures as well on leucocytes, hemoglobin und thrombocytes should be included if the authors want to speculate on possible side effects.
Author Response
Thanks for the reviewer’s comments and we addressed the corresponding questions in the attachment.
